# Bi-channel image registration and deep-learning segmentation (BIRDS) for efficient, versatile 3D mapping of mouse brain

Xuechun Wang[1†], Weilin Zeng[1†], Xiaodan Yang[2†], Yongsheng Zhang[3†*], Chunyu Fang[1†], Shaoqun Zeng[3], Yunyun Han[2*], Peng Fei[1*]

[1]School of Optical and Electronic Information- Wuhan National Laboratory for Optoelectronics, Huazhong University of Science and Technology, Wuhan, China; [2]School of Basic Medicine, Tongji Medical College, Huazhong University of Science and Technology, Wuhan, China; [3]Britton Chance Center for Biomedical Photonics, Wuhan National Laboratory for Optoelectronics, Huazhong University of Science and Technology, Wuhan, China

*For correspondence:
ys-zhang20@mails.tsinghua.edu.cn (YZ);
yhan@hust.edu.cn (YH);
feipeng@hust.edu.cn (PF)

[†]These authors contributed equally to this work

Competing interests: The authors declare that no competing interests exist.

**Abstract** We have developed an open-source software called bi-channel image registration and deep-learning segmentation (BIRDS) for the mapping and analysis of 3D microscopy data and applied this to the mouse brain. The BIRDS pipeline includes image preprocessing, bi-channel registration, automatic annotation, creation of a 3D digital frame, high-resolution visualization, and expandable quantitative analysis. This new bi-channel registration algorithm is adaptive to various types of whole-brain data from different microscopy platforms and shows dramatically improved registration accuracy. Additionally, as this platform combines registration with neural networks, its improved function relative to the other platforms lies in the fact that the registration procedure can readily provide training data for network construction, while the trained neural network can efficiently segment-incomplete/defective brain data that is otherwise difficult to register. Our software is thus optimized to enable either minute-timescale registration-based segmentation of cross-modality, whole-brain datasets or real-time inference-based image segmentation of various brain regions of interest. Jobs can be easily submitted and implemented via a Fiji plugin that can be adapted to most computing environments.

## Introduction

The mapping of the brain and neural circuits is currently a major endeavor in neuroscience and has great potential for facilitating an understanding of fundamental and pathological brain processes (*Alivisatos et al., 2012*; *Kandel et al., 2013*; *Zuo et al., 2014*). Large projects, including the Mouse Brain Architecture project (*Bohland et al., 2009*), the *Allen Mouse Brain Connectivity Atlas* (*Oh et al., 2014*), and the Mouse Connectome project, have mapped the mouse brain (*Zingg et al., 2014*) in terms of cell types, long-range connectivity patterns, and microcircuit connectivity. In addition to these large-scale collaborative efforts, an increasing number of laboratories are also developing independent, automated, or semi-automated frameworks for processing brain data obtained for specific projects (*Fürth et al., 2018*; *Ni et al., 2020*; *Niedworok et al., 2016*; *Renier et al., 2016*; *Wang et al., 2020a*; *Iqbal et al., 2019*). With the improvement of experimental methods for dissection of brain connectivity and function, development of a standardized and automated computational pipeline to map, analyze, visualize, and share brain data has become a major challenge to all brain connectivity mapping efforts (*Alivisatos et al., 2012*; *Fürth et al., 2018*). Thus, the

**eLife digest** Mapping all the cells and nerve connections in the mouse brain is a major goal of the neuroscience community, as this will provide new insights into how the brain works and what happens during disease. To achieve this, researchers must first capture three-dimensional images of the brain. These images are then processed using computational tools that can identify distinct anatomical features and cell types within the brain.

Various microscopy techniques are used to capture three-dimensional images of the brain. This has led to an increasing number of computational programs that can extract data from these images. However, these tools have been specifically designed for certain microscopy techniques. For example, some work on whole-brain datasets while others are built to analyze specific brain regions. Developing a more flexible, standardized method for annotating microscopy images of the brain would therefore enable researchers to analyze data more efficiently and compare results across experiments.

To this end, Wang, Zeng, Yang et al. have designed an open-source software program for extracting features from three-dimensional brain images which have been captured using different microscopes. Similar to other tools, the program uses an 'image registration' method that is able to recognize and annotate features in the brain. These tools, however, are limited to whole-brain datasets in which the complete anatomy of each feature must be present in order to be recognized by the software.

To overcome this, Wang et al. combined the image registration method with a deep-learning algorithm which uses pixels in the image to identify features in isolated regions of the brain. Although these neural networks do not require whole-brain images, they do need large datasets to 'learn' from. Therefore, the image registration method also benefits the neural network by providing a dataset of annotated features that the algorithm can train on.

Wang et al. showed that their software program, named BIRDS, could accurately recognize pixel-level brain features within imaging datasets of brain regions, as well as whole-brain images. The deep-learning algorithm could also adapt to analyze various types of imaging data from different microscopy platforms. This open-source software should make it easier for researchers to share, analyze and compare brain imaging datasets from different experiments.

implementation of an efficient and reliable method is fundamentally required for defining the accurate anatomical boundaries of brain structures, by which the anatomical positions of cells or neuronal connections can be determined to enable interpretation and comparison across experiments (*Renier et al., 2016*). The commonly used approach for automatic anatomical segmentation is to register an experimental image dataset within a standardized, fully segmented reference space, thus obtaining the anatomical segmentation for this set of experimental images (*Oh et al., 2014*; *Ni et al., 2020*; *Renier et al., 2016*; *Kim et al., 2015*; *Lein et al., 2007*). There are currently several registration-based high-throughput image frameworks for analyzing large-scale brain datasets (*Fürth et al., 2018*; *Ni et al., 2020*; *Niedworok et al., 2016*; *Renier et al., 2016*). Most of these frameworks require the user to set a few parameters based on the image intensity or graphics outlines or to completely convert the dataset into a framework-readable format to ensure the quality of the resulting segmentation. However, with the rapid development of sample labeling technology (*Lee et al., 2016*; *Richardson and Lichtman, 2015*; *Schwarz et al., 2015*) and high-resolution whole-brain microscopic imaging (*Economo et al., 2016*; *Gong et al., 2013*; *Nie et al., 2020*; *Liu et al., 2017*; *Li et al., 2010*), the heterogeneous and non-uniform characteristics of brain structures make it difficult to use traditional registration methods for registering datasets from different imaging platforms to a standard brain space with high accuracy. In this case, laborious visual inspection, followed by manual correction, is often required, which significantly reduces the productivity of these techniques. Therefore, the research community urgently needs a robust, comprehensive registration method that can extract a significant number of unique features from image data and provide accurate registration between different types of individual datasets.

Moreover, though registration-based methods can achieve full anatomical annotation in reference to a standard atlas for whole-brain datasets, their region-based 3D registration to a whole-brain atlas

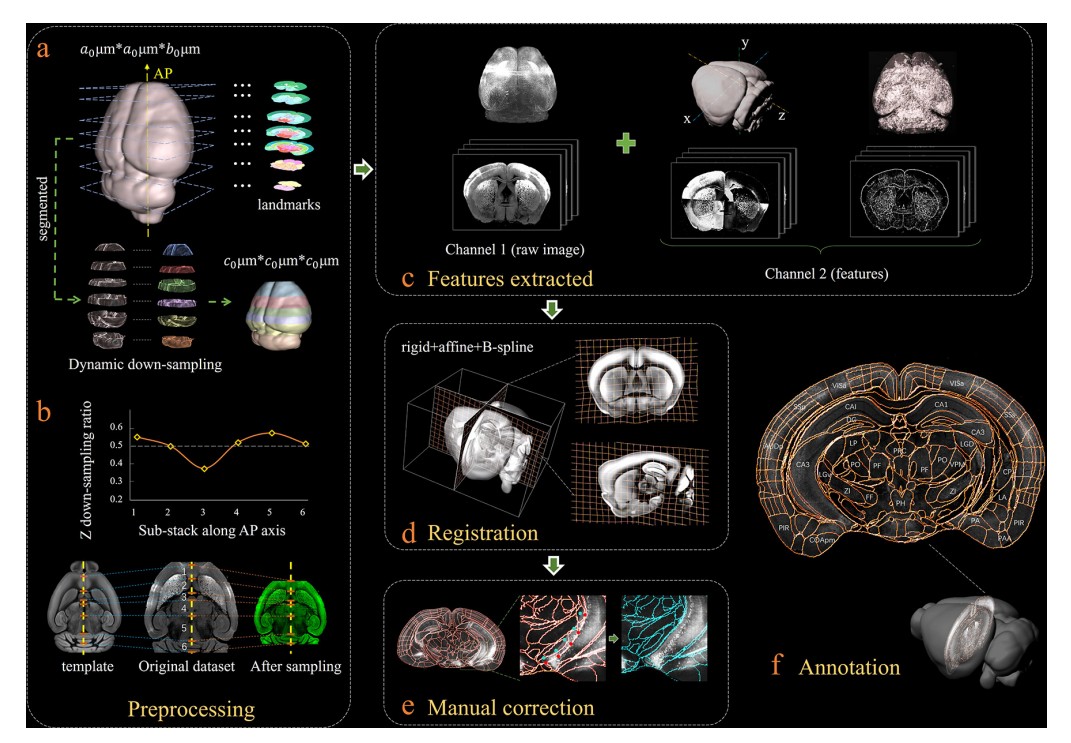

**Figure 1.** Bi-channel brain registration procedure. (**a**) Re-sampling of a raw 3D image into an isotropic low-resolution one, which has the same voxel size (20 μm) using an averaged Allen template image. The raw brain dataset was first subdivided into six sub-stacks along the AP axis according to landmarks identified in seven selected coronal planes (a1). Then an appropriate z re-sampling ratio, which was different for each slice, was applied to each sub-stack (a2, left) to finely adjust the depth of the stack in the down-sampled data (a2, right). This step roughly restored the deformation of non-uniformly morphed samples, thereby allowing the following registration with an Allen reference template. (**b**) Plot showing the variation of the down-sampling ratio applied to the six sub-stacks and comparison with the Allen template brain before and after the dynamic re-sampling showing the shape restoration effects of the this preprocessing step. (**c**) Additional feature channels containing a geometry and outline feature map extracted using grayscale reversal processing (left), as well as an edge and texture feature map extracted by a phase congruency algorithm (right). This feature channel was combined with a raw image channel for implementing our information-enriched bi-channel registration, which showed improved accuracy as compared to conventional single-channel registration solely based on raw images. (**d**) 3D view and anatomical sections (coronal and sagittal planes) of the registration results displayed in a grid deformed from an average Allen template. (**e**) Visual inspection and manual correction of automatically-registered results from an optically clarified brain, which showed obvious deformation. Using the GUI provided, this step could be readily operated by adjusting the interactive nodes in the annotation file (red points to light blue points). (**f**) A final atlas of an experimental brain image containing region segmentations and annotations.

The online version of this article includes the following figure supplement(s) for figure 1:

**Figure supplement 1.** Selection of featured coronal planes corresponded to the Allen Reference Atlas, for subdividing the entire brain volume into multiple sub-stacks along AP axis.

**Figure supplement 2.** Improved registration accuracy enabled by the abovementioned shape rectification preprocessing.

**Figure supplement 3.** Extraction of axes features for registration.

**Figure supplement 4.** Schematic of the multi-channel registration and annotation process.

lacks the flexibility to analyze incomplete brain datasets or those focused on a certain volume of interest (*Song and Song, 2018*), which is often the case in neuroscience research. Though some frameworks can register certain types of brain slabs that contain complete coronal outlines slice by slice (*Fürth et al., 2018*; *Song and Song, 2018*; *Ferrante and Paragios, 2017*), it remains very difficult to register a small brain block without obvious anatomical outlines. As neural networks have emerged as a technique of choice for image processing (*Long et al., 2015*; *Chen et al., 2018a*; *He et al., 2019*; *Zhang et al., 2019*), deep-learning-based brain mapping methods have also recently been reported to directly provide segmentation/annotation of primary regions for 3D brain datasets (*Iqbal et al., 2019*; *Akkus et al., 2017*; *Chen et al., 2018b*; *Milletari et al., 2017*; *de Brebisson and Montana, 2015*). Such deep-learning-based segmentation networks are efficient

in extracting pixel-level features and thus are not dependent on the presence of global features such as complete anatomical outlines, making them better suited for processing of incomplete brain data, as compared to registration-based methods. On the other hand, the establishment of these networks still relies on a sufficiently large training dataset, which is often laboriously registered, segmented, and annotated. Therefore, a combination of image registration and a neural network can possibly provide a synergistic improved analysis method and lead to more efficient and versatile brain mapping techniques.

Here, we provide an open-source software as a Fiji (*Schindelin et al., 2012*) plugin, termed bi-channel image registration and deep-learning segmentation (BIRDS), to support brain mapping efforts and to make it feasible to analyze, visualize, and share brain datasets. We developed BIRDS to allow investigators to quantify and spatially map 3D brain data in its own 3D digital space with reference to Allen CCFv3 (*Wang et al., 2020b*). This facilitates analysis in its native status at cellular level. The pipeline features: (1) A bi-channel registration algorithm integrating a feature map with raw image data for co-registration with significantly improved accuracy and (2) a mutually beneficial strategy in which the registration procedure can readily provide training data for a neural network, while this network can efficiently segment incomplete brain data that is otherwise difficult to register with a standardized atlas. The whole computational framework is designed to be robust and flexible, allowing its application to a wide variety of imaging systems (e.g., epifluorescent microscopy or light-sheet microscopy) and labeling approaches (e.g., fluorescent proteins, immunohistochemistry, and in situ hybridization). The BIRDS pipeline offers a complete set of tools, including image preprocessing, feature-based registration and annotation, visualization of digital maps and quantitative analysis via a link with Imaris, and a neural network segmentation algorithm that allows efficient processing of incomplete brain data. We further demonstrate how BIRDS can be employed for fully automatic mapping of various brain structures and integration of multidimensional anatomical neuronal labeling datasets. The whole pipeline has been packaged into a Fiji plugin, with step-by-step tutorials that permit rapid implementation of this plugin in a standard laboratory computing environment.

## Results

### Bi-channel image registration with improved accuracy

*Figure 1* shows our bi-channel registration procedure, which registers experimental whole-brain images using a standardized Allen Institute mouse brain average template, and then provides segmentations and annotations from CCFv3 for experimental data. The raw high-resolution 3D images (1 × 1 × 10 μm$^3$ per voxel), obtained by serial two-photon tomography (STPT, see Materials and methods), were first down-sampled into isotropic low-resolution data with a 20 μm voxel size identical to an averaged Allen template image (*Figure 1a*). The re-sampling ratios along the x (lateral-medial axis), y (dorsal-ventral axis), and z (anterior-posterior, AP axis) axes were thus 0.05, 0.05 and 0.5, respectively. It should be noted that, in addition to the individual differences, the preparation/mounting steps can also cause non-uniform deformation of samples, thereby posing extra challenges to the precise registration of experimental image to an averaged template (*Figure 1b*, original dataset). To mitigate this non-uniform deformation issue before registration, we applied a dynamic re-sampling ratio rather than using a fixed value of 0.5 to the z reslicing. We first subdivided the entire image stack into multiple sub-stacks (n = 6 in our demonstration, *Figure 1a*) according to seven selected landmark planes (*Figure 1a*, *Figure 1—figure supplement 1*). Then we applied a dynamic z re-sampling ratio calculated corresponding to the positions of the landmark planes in the Allen template and sample data (varying from ~0.35 to 0.55) to each sub-stack, to finely compress (<0.5) or stretch (>0.5) the z depth of the sub-stacks, thereby better matching the depth of each sub-stack to the Allen template brain and rectifying the deformation along the AP axis (*Figure 1a*, Materials and methods). The rectified whole-brain stack assembled by these dynamically re-sampled sub-stacks showed higher original similarity to the Allen template brain as compared to a raw experimental image stack (*Figure 1b*). The implementation of such a preprocessing step was beneficial for the better alignment of non-uniformly morphed brain data to a standardized template (*Figure 1—figure supplement 2*). After data preprocessing, we applied a feature-based iterative registration using the Allen reference images to the preprocessed experimental images. We note that previous

registration methods were vulnerable to inadequate alignment accuracy (*Niedworok et al., 2016*; *Renier et al., 2016*; *Goubran et al., 2019*), which was associated with inadequate registration information provided by merely using the raw background image data. To address this issue, in addition to the primary channel containing the background images of each sample and template brains, we further generated an assistant channel to augment the image registration and enhance the accuracy. First, we used a phase congruency (PC) algorithm (*Kovesi, 2019*) to extract the high-contrast edge and texture information from both the experimental and template brain images based on their relatively fixed anatomy features (*Figure 1c*, Materials and methods). Then, we obtained the geometry features of both brains along their lateral–medial, dorsal–ventral, and anterior–posterior axes with enhanced axial mutual information (MI) extracted using a grayscale reversal processing (*Maes et al., 1997*; *Thévenaz and Unser, 2000*) (*Figure 1c*, *Figure 1—figure supplement 3*, Materials and methods). Finally, the primary channel containing raw brain images, in conjunction with the assistant channel containing the texture and geometry maps of brains, were included in the registration procedure to fulfill an information-augmented bi-channel registration requirement (*Figure 1—figure supplement 4*), which was verified to show notably better registration accuracy as compared to conventional single-channel registration methods (aMAP [*Niedworok et al., 2016*], ClearMap [*Renier et al., 2016*], and MIRACL [*Goubran et al., 2019*]). During registration, through an iterative optimization of the transformation from an averaged Allen brain template to the experimental data, the MI gradually reached its maximum when the inverse grayscale images, PC images, and the raw images were finally geometrically aligned (*Figure 1d*). The displacement was presented in a grid form to illustrate the non-linear deformation effects. The geometry wrapping parameters obtained from the registration process were then applied to the Allen annotation file to generate a transformed version specifically for experimental data (*Figure 1—figure supplement 4*). Our dual-channel registration achieved fully automated registration/annotation at sufficiently high accuracy when processing STPT experimental data of an intact brain (*Han et al., 2018*). As for low-quality or highly deformed brain data (e.g., clarified brain with obvious shrinkage), though the registration accuracy of our method was accordingly reduced, our method still quite obviously surpassed other methods (*Figure 2*). For such challenging data types, we also developed an interactive graphic user interface (GUI) to readily permit manual correction of the visible inaccuracies in the annotation file, through finely tuning the selected corresponding points (*Figure 1e*). Finally, an accurate 3D annotation could be generated and applied to experimental data, either fully automatically (STPT data) or after mild manual correction (light-sheet fluorescence microscopy [LSFM] data of clarified brain), as shown in *Figure 1f*.

## Comparison with conventional single-channel-based registration methods

Next, we merged our experimental brain image with a registered annotation file to generate a 3D annotated image and quantitatively compared its registration accuracy with aMAP, ClearMap, and MIRACL results. We made comparisons of both STPT data from intact brains that contained only minor deformations (*Figure 2a*) and LSFM data from clarified brains (u-DISCO) that showed obvious shrinkage (*Figure 2b*). It should be noted here that the annotated results of either previous single-channel methods or our bi-channel method were all using automatic registration without any manual correction applied, and the averaged manual annotations by our experienced researchers served as a ground truth for quantitative comparisons. It was visually obvious that, as compared to the other three methods (green: aMAP; red: ClearMap; and blue: MIRACL in *Figure 2a,b*), the Allen annotation files transformed and registered by our BIRDS method (yellow in *Figure 2a,b*) were far better aligned with both STPT (as shown in VISC, CENT, AL, and PAL regions, *Figure 2a*) and LSFM (as shown in HPF, CB, VIS, and COA regions, *Figure 2b*) images. Furthermore, we manually labeled 10 3D fiducial points of interest (POIs) across the registered Allen template images together with their corresponding experimental images (*Figure 2c*) and then measured the error distances between the paired anatomical landmarks in the two datasets, so that the registration accuracy by each registration method could be quantitatively evaluated (*Figure 2—figure supplement 1*). As shown in *Figure 2d*, the error distance distributions of POIs in five brains (two STPT + three LSFM) registered by the abovementioned four methods were then quantified, showing the smallest median error distance (MED) was obtained using our method for all five brains (*Supplementary file 3*). In two different sets of STPT data, only our BIRDS method could provide an MED below 100 μm (~80 μm, n = 2),

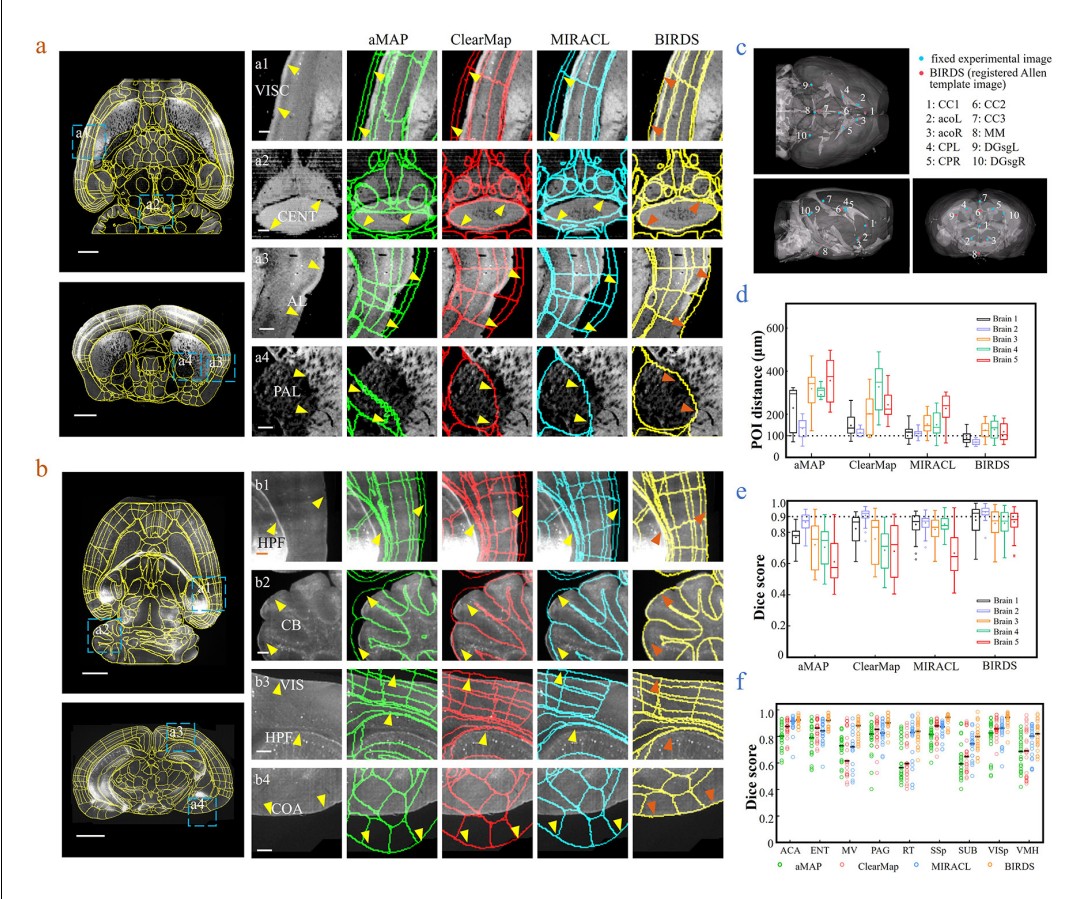

**Figure 2.** Comparison of BIRDS with conventional single-channel registration methods. (**a**) Comparative registration accuracy (STPT data from an intact brain) using four different registration methods, aMAP, ClearMap, MIRACL, and BIRDS. (**b**) Comparative registration accuracy (LSFM data from clarified brain) using four methods. Magnified views of four regions of interest (a1–a4, b1–b4, blue boxes) selected from the horizontal (left, top) and coronal planes (left, bottom) are shown in the right four columns, with 3D detail for the registration/annotation accuracy for each method. All comparative annotation results were directly output from respective programs without manual correction. Scale bar, 1 mm (whole-brain view) and 250 μm (magnified view). (**c**) Ten groups of 3D fiducial points of interest (POIs) manually identified across the 3D space of whole brains. The blue and red points belong to the fixed experimental images and the registered Allen template images, respectively. The ten POIs were selected from the following landmarks: *POIs: cc1*: corpus callosum, midline; *acoL, acoR*: anterior commisure, olfactory limb; *CPL, CPR*: Caudoputamen, Striatum dorsal region; *cc2*: corpus callosum, midline; *cc3*: corpus callosum, midline; *MM*: medial mammillary nucleus, midline; *DGsgL, DGsgR*: dentate gyrus, granule cell layer. The registration error by each method could be thereby quantified through measuring the Euclidean distance between each pair of POIs in the experimental image and template image. (**d**) Box diagram comparing the POI distances of five brains registered by the four methods. Brains 1, 2: STPT images from two intact brains. Brain one is also shown in (**a**). Brains 3, 4, and 5: LSFM images from three clarified brains (u-DISCO) that showed significant deformations. Brain five is also shown in (**b**). The median error distance of 50 pairs of POIs in the five brains registered by BIRDS was ~104 μm, as compared to ~292 μm for aMAP, ~204 μm for ClearMap, and ~151 μm for MIRACL. (**e, f**) Comparative plot of Dice scores in nine registered regions of the five brains. The results were grouped by brain in (**e**) and region in (**f**). The calculation was implemented at the single nuclei level. When the results were analyzed by brain, BIRDS surpassed the other three methods most clearly using LSFM dataset #5, with a 0.881 median Dice score as compared to 0.574 from aMAP, 0.72 from ClearMap, and 0.645 from MIRACL. At the same time, all the methods performed well on STPT dataset #2, with a median Dice score of 0.874 from aMAP, 0.92 from ClearMap, 0.872 from MIRACL, and 0.933 from BIRDS. When the results were compared using nine functional regions, the median values acquired by BIRDS were also higher than the other three methods. Even the lowest median Dice score by our method was still 0.799 (indicated by black line), which was notably higher than 0.566 by aMAP, 0.596 by ClearMap, and 0.722 by MIRACL, respectively.

The online version of this article includes the following source data and figure supplement(s) for figure 2:

**Source data 1.** Source data file for *Figure 2*.
**Figure supplement 1.** Selection of 10 fiducial points of interest (POIs) for measuring the error distance between registered brains.
**Figure supplement 2.** Comparison between BIRDS and conventional single-channel registration.
**Figure supplement 3.** Comparative accuracy analysis of BIRDS and single-channel registration.
**Figure supplement 3—source data 1.** Source data file for *Figure 2—figure supplement 3*.
**Figure supplement 4.** Dice score comparison of nine regions in five brains registered by four registration tools: aMAP, ClearMap, MIRACL, and our BIRDS.
*Figure 2 continued on next page*

*Figure 2 continued*

**Figure supplement 4—source data 1.** Source data file for *Figure 2—figure supplement 4*.

and this value slightly increased to ~120 µm for LSFM data (n = 3), but was still smaller than all the results obtained using the other three methods (aMAP, ~342 µm, n = 3; ClearMap, ~258 µm, n = 3; and MIRACL, ~175 µm, n = 3). Moreover, the Dice scores (*Dice, 1945*), defined as a similarity scale function used to calculate the similarity of two samples, for each method were also calculated at the nucleus precision level based on nine functional regions in the five brains. The comparative results were then grouped by brain and region, as shown in *Figure 2e,f*, respectively. The highest Dice scores with an average median value of >0.89 (*Supplementary file 3*, calculated for five brains, 0.75, 0.81, and 0.81 for aMAP, ClearMap, and MIRACL) or >0.88 (*Supplementary file 3*, calculated using nine regions, 0.74, 0.77, and 0.84 for aMAP, ClearMap, and MIRACL, respectively) were obtained by BIRDS, further confirming the superior registration accuracy of our method. Through a comparative Wilcoxon test, our results were demonstrated to be superior to the other three methods (providing larger Dice scores) with a p value < 0.05 calculated either by brain or by region. More detailed comparisons of registration accuracies can be found in *Figure 2—figure supplements 2–4*.

## Whole-brain digital map identifying the distributions of labeled neurons and axon projections

A 3D digital map (CCFv3) based on the abovementioned bi-channel registration was generated to support automatic annotation, analysis, and visualization of neurons in a whole mouse brain (see Materials and methods). The framework thus enabled large-scale mapping of neuronal connectivity and activity to reveal the architecture and function of brain circuits. Here, we demonstrated how the BIRDS pipeline visualizes and quantifies single-neuron projection patterns obtained by STPT imaging. A mouse brain containing six GFP-labeled layer-2/3 neurons in the right visual cortex was imaged with STPT at $1 \times 1 \times 10$ µm$^3$ resolution (*Han et al., 2018*). After applying the BIRDS procedure to this STPT image stack, we generated a 3D map of this brain (*Figure 3a*). An interactive hierarchal tree of brain regions in the software interface allowed navigation through the corresponding selected-and-highlighted brain regions with its annotation information (*Figure 3b*, *Video 1*). Through linking with Imaris, we visualized and traced each fluorescently labeled neuronal cell (n = 5) using the filament module of Imaris across the 3D space of the entire brain (*Figure 3c*, Materials and methods, *Video 2*). The BIRD software can also apply reverse transformation to a raw image stack to generate a standard template-like rendered 3D map, including both traced axonal projections and selected whole-brain structures, which faithfully captures true 3D axonal arborization patterns and anatomical locations, as shown in *Figure 3d*. This software can also quantify the lengths and arborizations of traced axons according to the segmentation of the 3D digital map generated using the BIRDS pipeline (*Figure 3e*).

BIRDS can be linked to Imaris to perform automated cell counting with higher efficiency and accuracy (Materials and methods). Here, we demonstrate it with an example brain where neurons were retrogradely labeled by CAV-mCherry injected to the right striatum and imaged by STPT at $1 \times 1 \times 10$ µm$^3$ resolution (*Han et al., 2018*). The whole-brain image stacks were first processed by BIRDS to generate a 3D annotation map. Two of the example segregated brain areas (STR and BS) are outlined in the left panel of *Figure 4a*. The annotation map and the raw image stack were then transferred to Imaris, which processed the images within each segregated area independently. Imaris calculated the local image statistics for cell recognition only using the image stack within each segregated area; therefore, it fit the dynamic range of the local images to achieve better results, as shown in the middle column in the right panel of *Figure 4a*. In contrast, the conventional Imaris automated cell counting program processed the whole-brain image stack at once to calculate the global cell recognition parameters for every brain area, which easily resulted in false positive or false negative counts in brain areas where the labeling signal was too strong or too weak compared to the global signal, as demonstrated in the STR and SB in the right column of the right panels of *Figure 4a*, respectively. The BIRDS–Imaris program could perform automated cell counting for each brain area and reconstructed them over the entire brain. The 3D model of the brain-wise distribution of labeled striatum-projecting neurons was visualized using the BIRDS–Imaris program as a 3D rendered brain

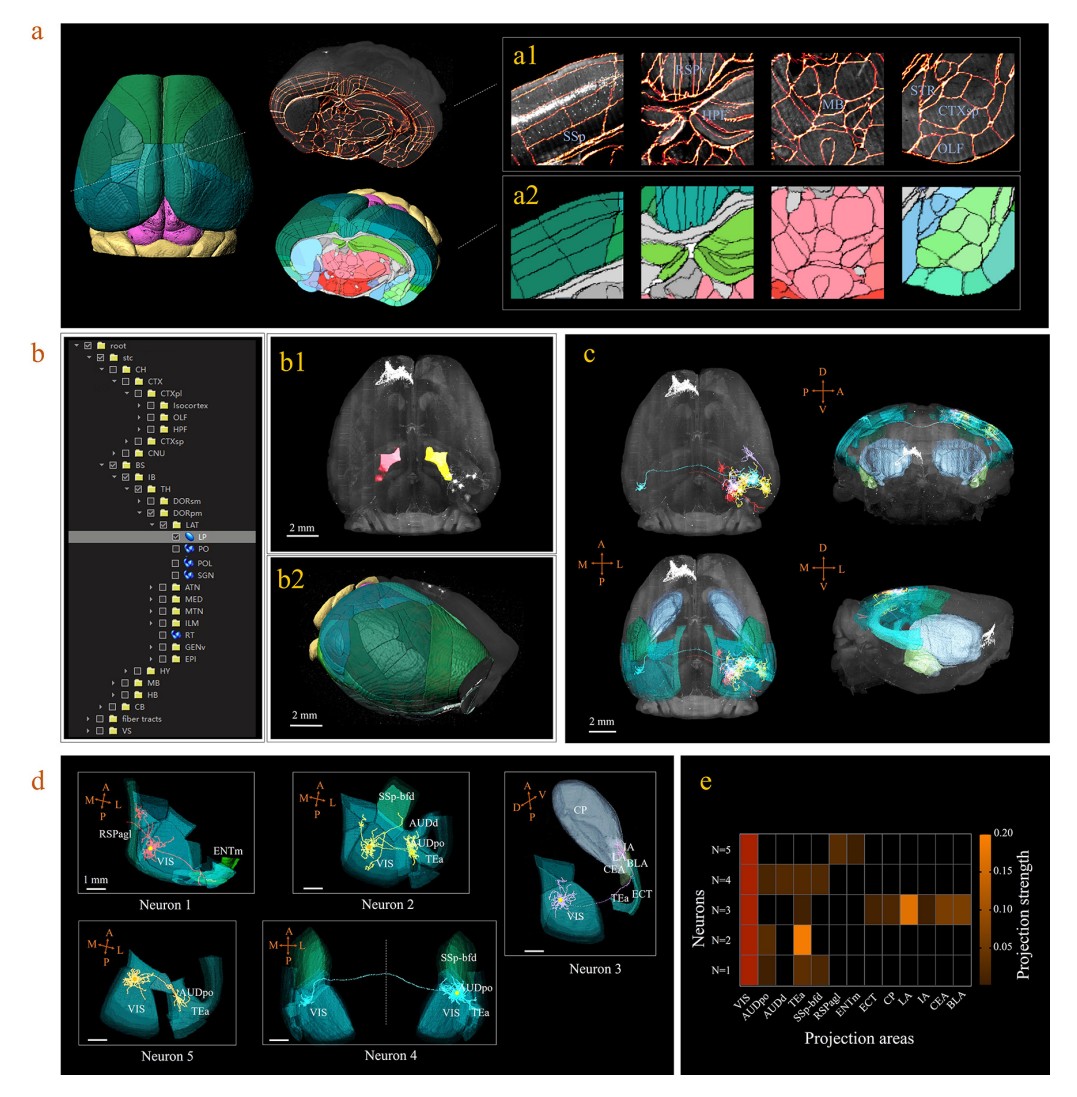

**Figure 3.** 3D digital atlas of a whole brain for visualization and quantitative analysis of inter-areal neuronal projections. (**a**) Rendered 3D digital atlas of a whole brain (a2, pseudo color), which was generated from registered template and annotation files (a1, overlay of annotation mask and image data). (**b**) Interactive hierarchical tree shown as a sidebar menu in the BIRDS program, indexing the name of brain regions annotated in CCFv3. Clicking on any annotation name in the side bar of the hierarchal tree highlights the corresponding structure in the 3D brain map (b1, b2), and vice versa. For example, brain region LP was highlighted in the space after its name was chosen in the menu (b1). 3D rendering of an individual brain after applying a deformation field in reverse to a whole brain surface mask. The left side of the brain displays the 3D digital atlas (CCFv3, colored part in b2), while the right side of the brain is displayed in its original form (grayscale part in b2). (**c**) The distribution of axonal projections from five single neurons in 3D map space. The color-rendered space shown in horizontal, sagittal, and coronal views highlights multiple areas in the telencephalon, anterior cingulate cortex, striatum, and amygdala, which are all potential target areas of layer-2/3 neuron projections. (**d**) The traced axons of five selected neurons (n = 5) are shown. ENTm, entorhinal area, medial part, dorsal zone; RSPagl, retrosplenial area, lateral agranular part; VIS, visual areas; SSp-bfd, primary somatosensory area, barrel field; AUDd, dorsal auditory area; AUDpo, posterior auditory area; TEa, temporal association areas; CP, caudoputamen; IA, intercalated amygdalar nucleus; LA, lateral amygdalar nucleus; BLA, basolateral amygdalar nucleus; CEA, central amygdalar nucleus; ECT, ectorhinal area. (**e**) Quantification of the projection strength across the targeting areas of five GFP-labeled neurons. The color codes reflect the projection strengths of each neuron, as defined as axon length per target area, normalized to the axon length in VIS.

The online version of this article includes the following source data for figure 3:

**Source data 1.** Source data file for *Figure 3e*.

image and projection views from three axes in *Figure 4b*. The BIRDS program could calculate the volume of each segregated region according to the 3D segregation map and the density of labeled cells across the brain as shown in *Figure 4c*. Meanwhile, manual cell counting was also performed

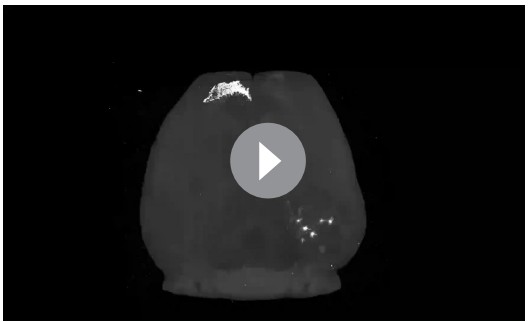

**Video 1.** Displays the 3D digital atlas.
https://elifesciences.org/articles/63455#video1

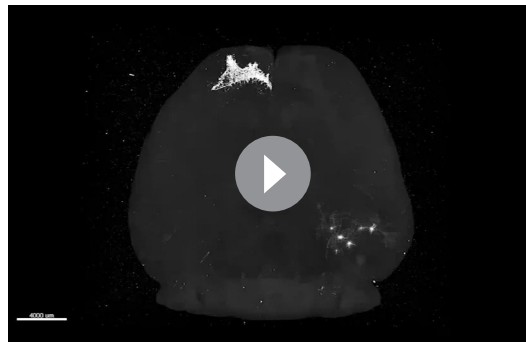

**Video 2.** Shows the arborization of 5 neurons in 3D map space.
https://elifesciences.org/articles/63455#video2

with every one out of four sections using an ImageJ plugin (*Figure 4d*). Compared to conventional Imaris results, our BIRDS–Imaris results were more consistent with a manual one, especially for brain regions where the fluorescent signal was at the high or low end of the dynamic range (BS and STR, *Figure 4e*). Thanks to the 3D digital map generated by the BIRDS pipeline, BIRDS–Imaris can process each segmented brain area separately, namely calculating the parameters for the cell recognition algorithm using local image statistics instead of processing the whole-brain image stack at once. Such a segmented cell counting strategy is much less demanding on computation resources, and moreover, it is optimized for each brain area to solve the problem that the same globe cell recognition parameter works poorly in certain brain regions with signal intensity at either of the two extreme ends of the dynamic range of the entire brain.

## Inference-based segmentation of incomplete brain datasets using a deep-learning procedure

In practice, acquired brain datasets are often incomplete, due to researcher's particular interest in specific brain regions, or limited imaging conditions. The registration of such incomplete brain datasets to an Allen template is often difficult due to the lack of sufficient morphology information for comparison of both datasets. To overcome this limitation, we further introduced a deep neural network (DNN)-based method for efficient segmentation/annotation of incomplete brain sections with minimal human supervision. Herein, we optimized a Deeplab V3+ network, which was based on an encoding-decoding structure, for our deep-learning implementation (*Figure 5a*). The input images passed through a series of feature processing stages in the network, with pixels being allocated, classified, and segmented into brain regions. It should be noted that the training of a neural network fundamentally requires a sufficiently large dataset containing various incomplete brain blocks which have been well segmented. Benefiting from our efficient BIRDS method, we could readily obtain a large number of such labeled datasets through cropping processed whole brains and without experiencing time-consuming manual annotation. Various types of incomplete brains, as shown in *Figure 5b*, were hereby generated and sent to our DNN for iterative training, after which the skilled network could directly infer the segmentations/annotations for new modes of incomplete brain images (Materials and methods). Next, we validated the network performance on three different modes of input brain images cropped from the registered whole-brain dataset (STPT). The DNN successfully inferred annotation results for a cropped hemisphere, irregular cut of hemisphere, and a randomly cropped volume, as shown in *Figure 5c–e*, respectively. The inferred annotations (red lines) were found to be highly similar to the registered annotation results (green lines) in all three types of incomplete data. To further quantify the inference accuracy, the Dice scores of the network-segmented regions were also calculated by comparing the network outputs to ground truth, which was the registration results after visual inspection and correction (*Figure 5—figure supplement 1*). The averaged median Dice scores for the individual sub-regions in the hemisphere, irregular cut of hemisphere, and random volumes were 0.86, 0.87, and 0.87, respectively, showing a sufficiently high inference accuracy in most of brain regions, such as the isocortex, HPF, OLF, or STR. It is worth

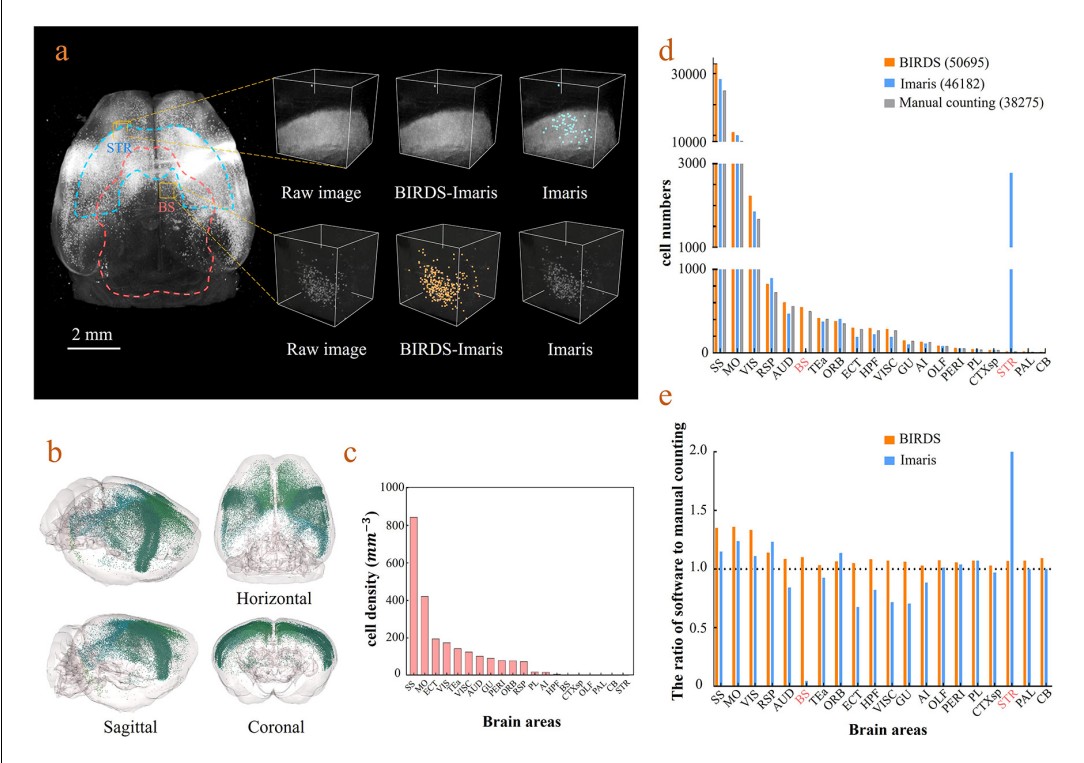

**Figure 4.** Cell-type-specific counting and comparison between different cell counting methods. (**a**) Cell counting of retrogradely labeled striatum-projecting cells. We selected two volumes (1 × 1 × 1 mm³) from SS and VIS areas, respectively, to show the difference in cell density and the quantitative results by BIRDS–Imaris versus conventional Imaris. Here, a separate quantification parameters set for different brain areas in the BIRD–Imaris procedure lead to obviously more accurate counting results. Scale bar, 2 mm. (**b**) 3D-rendered images of labeled cells in the whole brain space, shown in horizontal, sagittal, and coronal views. The color rendering of the cell bodies was in accordance with CCFv3, and the cells were mainly distributed in the Isocortex (darker hue). (**c**) The cell density calculated for 20 brain areas. The cell densities of MO and SS were highest (MO = 421.80 mm⁻³; SS = 844.71 mm⁻³) among all areas. GU, gustatory areas; TEa, temporal association areas; AI, agranular insular area; PL, prelimbic area; PERI, perirhinal area; RSP, retrosplenial area; ECT, ectorhinal area; ORB, orbital area; VISC, visceral area; VIS, visual areas; MO, somatomotor areas; SS, somatosensory areas; AUD, auditory areas; HPF, hippocampal formation; OLF, olfactory areas; CTXsp, cortical subplate; STR, striatum; PAL, pallidum; BS, brainstem; CB, cerebellum. (**d**) Comparison of the cell numbers from three different counting methods, BIRDS, Imaris (3D whole brain directly), and manual counting (2D slice by slice for a whole brain). (**e**) The cell counting accuracy using BIRDS–Imaris (orange) and conventional Imaris methods (blue), relative to manual counting. Besides the highly divergent accuracy for the 20 regions, the counting results by conventional Imaris in STR and BS regions were especially inaccurate.

The online version of this article includes the following source data for figure 4:

**Source data 1.** Source data file for *Figure 4*.

noting that the performance of our network for segmentation using PAL, MBsta, P-sen regions remained sub-optimal (Dice score 0.78–0.8), due to their lack of obvious borders, and large structural variations across planes (*Figure 5—figure supplement 1*). Finally, we applied our network inferences to generate 3D atlases for these three incomplete brains, while segmenting the hemisphere into 18 regions such as the Isocortex, HPF, OLF, CTXsp, STR, PAL, CB, DORpm, DORsm, HY, MBsen, MBmot, MBsta, P-sen, P-mot, P-sat, MY-sen, and MY-mot, while we processed an irregular cut of half the telencephalon into 10 regions as Isocortex, HPF, OLF, CTXsp, STR, PAL, DORpm, DORsm, and HY, MY-mot, and the random volume into seven regions, defined as the Isocortex, HPF, STR, PAL, DORpm, DORsm, and HY (*Figure 5f,g,h*). Therefore, our DNN performed reasonably well even if the brain was highly incomplete. Furthermore, it could achieve second-level fine segmentation within a small brain region of interest. For example, we successfully segmented the hippocampus (CA1, CA2, CA3, and DG), as shown in *Figure 5—figure supplement 2*. Such a unique capability of our DNN was possibly derived from the detection of pixel-level features rather than regions, and thereby substantially strengthened the robustness of our hybrid BIRDS method over conventional brain registration techniques when the data is highly incomplete/defective. More

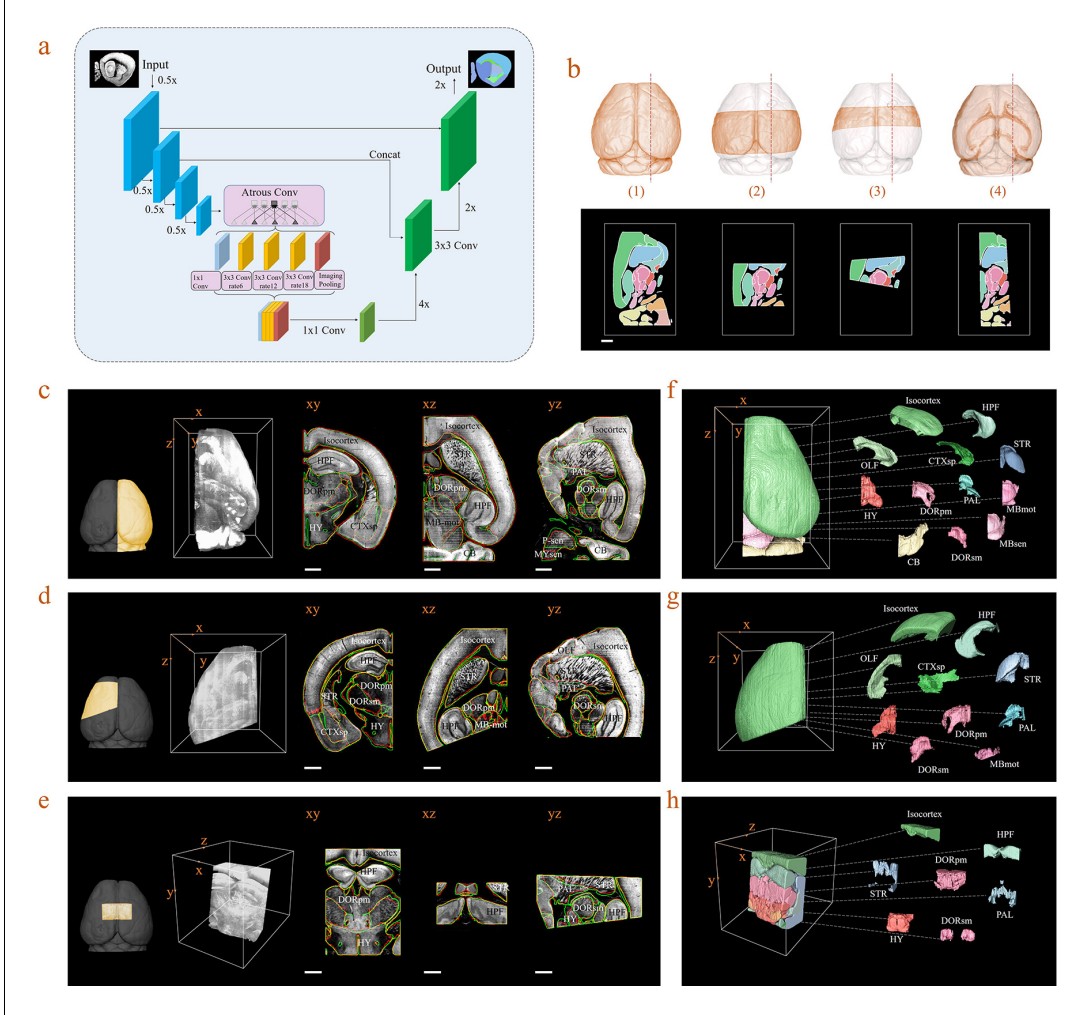

**Figure 5.** Inference-based network segmentation of incomplete brain data. (**a**) The deep neural network architecture for directly inferring brain segmentation without required registration. The training datasets contained various types of incomplete brain images, which were cropped from annotated whole-brain datasets created using our bi-channel registration beforehand. (**b**) Four models of incomplete brain datasets for network training: whole brain (1), a large portion of telencephalon (2), a small portion of telencephalon (3), and a horizontal slab of whole brain (4). Scale bar, 1 mm. (**c–e**) The inference-based segmentation results for three new modes of incomplete brain images, defined as the right hemisphere (**c**), an irregular cut of half the telencephalon (**d**), and a randomly cropped volume (**e**). The annotated sub-regions are shown in the x-y, x-z, and y-z planes, with the Isocortex, HPF, OLF, CTXsp, STR, PAL, CB, DORpm, DORsm, HY, MBsen, MBmot, MBsta, P-sen, P-mot, P-sat, MY-sen, and MY-mot for right hemisphere, Isocortex, HPF, OLF, CTXsp, STR, PAL, DORpm, DORsm, HY, and MY-mot for the irregular cut of half the telencephalon, and Isocortex, HPF, STR, PAL, DORpm, DORsm, and HY for the random volume. Scale bar, 1 mm. (**f–h**) Corresponding 3D atlases generated for these three incomplete brains.

The online version of this article includes the following source data and figure supplement(s) for figure 5:

**Source data 1.** Source data file for *Figure 5*.

**Figure supplement 1.** Dice scores of the DNN-segmented regions in three new types of incomplete brains.

**Figure supplement 1—source data 1.** Source data file for *Figure 5—figure supplement 1*.

**Figure supplement 2.** Performance of DNN inference for segmenting fine sub-regions specifically in the hippocampus.

**Figure supplement 3.** Accuracy comparison between DNN-based and registration-based brain segmentation.

**Figure supplement 4.** Dice scores of the DNN-segmented regions in abovementioned four types of brains.

**Figure supplement 4—source data 1.** Source data file for *Figure 5—figure supplement 4*.

**Figure supplement 5.** Comparison of the performance of DNN inference by three deep learning-based brainregion segmentation techniques, QuickNAT, SeBRe and our BIRDS (DNN).

detailed performance comparisons between our DNN-based inference and other methods are shown in *Figure 5—figure supplements 1–5*.

## Discussion

In summary, we demonstrate a bi-channel image registration method, in conjunction with a deep-learning framework, to readily provide accuracy-improved anatomical segmentation for whole mouse brain in reference to an Allen average template, and direct segmentation inference for incomplete brain datasets, which were otherwise not easily registered to standardized whole-brain space. The addition of a brain feature channel to the registration process greatly improved the accuracy of automatically registering individual whole-brain data with a standardized Allen average template. It should be noted that the registration was based on two-photon template images provided by Allen CCF, so it is currently limited to using on like-imaged brains, for example, brains imaged using wide-field, confocal, or light-sheet microscopes, etc. For processing various incomplete brain datasets, which were challenging for registration-based methods while remaining very common in neuroscience research, we applied our deep neural network to rapidly infer segmentations. The sufficiently accurate results shown using different types of incomplete data verify the advances of network segmentation. Though a full annotation using a neural network is currently too computationally demanding as compared to registration-based segmentation, it is undoubtedly a good complement to registration-based segmentation. Therefore, in our hybrid BIRDS pipeline, the DNN inference greatly reinforced the inefficient side of registration, while the registration also readily provided high-quality training data for our DNN. We believe such a synergistic effect in our method could provide a paradigm shift for enabling robust and efficient 3D image segmentation/annotation for biology research. With the unceasing development of deep learning, we envision that network-based segmentation will play an increasingly important role in new pipelines. A variety of applications, such as tracing of long-distance neuronal projections and parallel counting of cell populations in different brain regions, was also enabled as a result of our efficient brain mapping. The BIRDS pipeline is now fully open source and also has been packaged into a Fiji plugin to facilitate biological researchers. We sincerely expect that the BIRDS method can immediately allow new insights using current brain mapping techniques, and thus further push the resolution and scale limits in future explorations of brain space.

## Materials and methods

### Acquisition of STPT image dataset

Brains 1 and 2 were obtained with STPT, and each dataset encompassed ~180 Gigavoxels, for example, 11,980 × 7540 × 1075 in Dataset 1, with a voxel size of $1 \times 1 \times 10 \ \mu m^3$. The procedure of sample preparation and imaging acquisition were described in *Han et al., 2018*. Briefly, the adult C57BL/6 mouse (RRID:IMSR_JAX:000664) was anesthetized, and craniotomy was performed on top of the right visual cortex. Individuals neuronal axons were labeled with plasmid DNA (pCAG-eGFP [Addgene accession 11150]) by two-photon microscopy-guided single-cell electroporation, and the brain was fixed by cardioperfusion of 4% paraformaldehyde 8 days later. Striatum-projecting neurons were labeled by stereotactically injecting PRV-cre into the right striatum of tdTomato reporter mice (Ai14, JAX), and the brain was fixed cardioperfusion 30 days later. The brains were embedded in 5% oxidized agarose and imaged with a commercial STPT (TissueVision, USA) excited at 940 nm. Coronally, the brain was optically scanned every 10 μm at 1 μm/pixel without averaging and physically sectioned every 50 μm. The power of excitation laser was adjusted to compensate the depth of optical sections.

### Acquisition of LSFM images dataset

Brains 3, 4, and 5 were obtained with LSFM and each dataset encompassed ~700 Gigavoxels (~10,000 × 8000 × 5000), with an isotropic voxel size of $1 \ \mu m^3$. Brain tissues of eight-week-old Thy-GFP-M mice (RRID:IMSR_JAX:007788) were first clarified with u-DISCO protocol (*Pan et al., 2016*) before imaging. Brains 3 and 4 were acquired using a custom-built Bessel plane illumination microscope, a type of LSFM modality employing non-diffraction thin Bessel light-sheet. Brain 5 was whole-

brain 3D image of a Thy-GFP-M mice acquired using a lab-built selective plane illumination microscope (*Nie et al., 2020*), another LSFM modality combining Gaussian light-sheet with multi-view image acquisition/fusion.

## Implementation of Bi-channel registration

### Preprocessing of raw data

First, we developed an interactive GUI in the Fiji plugin (RRID:SCR_002285) to correspond the coronal planes in the Allen Reference Atlas (ARA) (132 planes, 100 μm interval) with those in the experimental 3D image stack (e.g., 1075 planes with 10 μm stepsize in Dataset 1). As shown in *Figure 1—figure supplement 1*, seven coronal planes, from the anterior bulbus olfactorius to the posterior cerebellum, were identified across the entire brain, with their number of layers being recorded as $a_i$ in template atlas and $b_i$ in the raw image stack. Therefore, k sub-stacks ($k = [1, 6], k \in N$) was defined by these seven planes (*Figure 1b*). According to the ratio of step size between the ARA and its template image stack (100 μm–20 μm), we also obtained the number of layers of the selected planes in the template image as $c_i = 5a_i - 2$. The reslicing ratio of the kth sub-stack sandwiched by every two planes ($a_k$ to $a_{k+1}$) was then calculated by: $lk = \frac{b_{k+1}-bk}{c_{k+1}-ck}$. Each $l_k$ was applied to the kth sub-stack to obtain the resliced version of the sub-stack. Finally, the six resliced sub-stacks together formed a complete image stack of whole brain (20 μm stepsize), which had a rectified shape more similar to the Allen average template image, as compared to the raw experimental data. According to the isotropic voxel size of 20 μm in the template, the lateral size of voxel in the resliced image stack was also adjusted from originally 1 μm to 20 μm with a uniform lateral down-sampling ratio of 20 applied to all the coronal planes. The computational cost of abovementioned data re-sampling operation was low, taking merely ~5 min for processing 180 GB raw STPT data on a Xeon workstation (E5-2630 V3 CPU).

### Features extraction and processing

We extracted feature information based on the purified signals of the image data with backgrounds filtrated. We realized this through calculating the threshold values of the signals and backgrounds using by Huang's fuzzy thresholding method (*Huang and Wang, 1995*) and removing the backgrounds according to the calculated thresholds. Then the feature information was detected using a PC algorithm, which was robust to the intensity change of signals, and could efficiently extract corners, lines, textures information from the image. Furthermore, when the images had relatively low contrast at the border, which was very common in our study, the edge information could be much better retained using PC detection. Finally, the pixel intensity in the generated PC feature map can be calculated by following formula:

$$\mathrm{E}(x) = \sum_{n} A_n [cos(\varphi_n(x) - \bar{\varphi}(x)) - |sin(\varphi_n(x) - \bar{\varphi}(x))|] \tag{1}$$

$$PC(x) = \frac{W(x)\lfloor|E(x)| - T\rfloor}{\sum_n A_n(x) + \varepsilon} \tag{2}$$

where $x$ is the angle vector of a pixel after Fourier transform of the image, $A_n$ is the local amplitude of the nth cosine component, $\varphi_n$ is the local phase, $\bar{\varphi}$ is the weighted average of phase, $\mathrm{E}(x)$ is the local energy, $W(x)$ is the filter band, $T$ is the noise threshold, and $\varepsilon$ is a small positive number (=0.01 in our practice) to prevent the denominator from leading to too large value of $PC(x)$.

### Bi-channel registration procedure

Image registration is fundamentally an iterative process optimized by a pre-designed cost function, which reasonably assesses the similarity between experimental and template datasets. Our bi-channel registration procedure is implemented based on the Elastix open-source program (Version 4.9.0) (*Shamonin et al., 2013*; *Klein et al., 2010*). Unlike conventional single-channel image registration, our method simultaneously registers all groups of input data using a single cost function defined as:

$$c\left(T_{\mu};I_F,I_M\right)=\frac{1}{\sum_{i=1}^{N}\omega_i}\sum_{i=1}^{N}\omega_i c\left(T_{\mu};I_F^i,I_M^i\right) \tag{3}$$

$$c\left(T_{\mu};I_F^i,I_M^i\right)=\frac{\sum_{m\in L_M}\sum_{f\in L_F}p\left(f,m;T_{\mu}\right)log_2\left(p_F(f)p_M\left(m;T_{\mu}\right)\right)}{\sum_{m\in L_M}\sum_{f\in L_F}p\left(f,m;T_{\mu}\right)log_2p\left(f,m;T_{\mu}\right)} \tag{4}$$

where $N$ represents the number of data groups and $\omega_i$ is the weighting parameter for each data group. Since we used primary channel containing raw image stack in conjunction with assistant channel containing geometry and texture feature maps for registration simultaneously, here $N=3$. $c\left(T_{\mu};I_F^i,I_M^i\right)$ is the cost function of each channel, where $I_F^i$ represents the fixed image (experimental data) and $I_M^i$ represents the moving image (template). $T_{\mu}$ denotes the deformation function of the registration model, with parameter μ being optimized during the iterative registration process. In *Equation 4*, $L_F$ and $L_M$ are two sets of regularly spaced intensity bin centers, $p$ is the discrete joint probability, and $p_F$ and $p_M$ are the marginal discrete probabilities of the fixed and moving image. Here we used rigid+affine+B-spline three-level model for the registration, with rigid and affine transformations mainly for aligning the overall orientation differences between the datasets, and B-spline model mainly for aligning the local geometry differences. B-spline places a regular grid of control points onto the images. These control points are movable during the registration and cause the surrounding image data to be transformed, thereby permitting the local, non-linear alignment of the image data. A stepsize of 30 pixels was set for the movement of control points in 3D space, and a five-level coarse-to-fine pyramidal registration was applied for achieving faster convergence. During the iterative optimization of $T_{\mu}$, we used gradient descent method to efficiently approach the optimal registration of the template images to the fixed experimental images. The solution of μ can be expressed as

$$\mu_{k+1}=\mu_k-l*\frac{\partial c\left(T_{\mu};I_F,I_M\right)}{\partial\mu} \tag{5}$$

where $l$ is the learning rate, which also means the stepsize of the gradient descent optimization. The transformation parameters obtained from the multi-channel registration were finally applied to the annotation files to generate the atlas /annotation for the whole-brain data.

## Visualization and quantification of brain-map results

### Generation of 3D digital map for whole brain

We obtained a low-resolution annotation (20 μm isotropic resolution) for entire mouse brain after registration. In order to generate a 3D digital framework based on the high-resolution raw image, $b,l$ recorded in the down-sampling step were used for resolution restoration. Annotation information is to distinguish different brain regions by pixel intensity. In order to generate a digital frame for quantitative analysis, we introduced Marching cubes algorithm (*Lorensen and Cline, 1987*) to generate 3D surface graphics, which is also, to generate the 3D digital maps. Then, through the programmable API link with Imaris (RRID:SCR_007370), we could readily visualize the 3D digital map and perform various quantification in Imaris. After registration and interpolation applied to the experimental data, a 3D digital map was visualized in Imaris (9.0.0) invoked by our program. Then neuron tracing and cell counting tasks could be performed in Imaris at native resolution (e.g., $1\times1\times10$ μm$^3$ for STPT data). During the neural tracing process, the brain regions where the selected neurons passed through could be three-dimensionally displayed under arbitrary view. Furthermore, the cell counting could be performed in parallel by simultaneously setting a number of kernel diameters and intensity thresholds for different segmented brain regions.

### Distance calculation

The Euclidean distance between one pair of landmarks shown in different image datasets (*Figure 2c*) indicates the registration error and can be calculated as:

$$\rho = \sqrt{(x_2 - x_1)^2 + (y_2 - y_1)^2 + (z_2 - z_1)^2} \tag{6}$$

## Calculation of dice scores

Dice score is the indicator for quantifying the accuracy of segmentation and can be calculated as:

$$Dice = \frac{2(A \cap B)}{A + B} \tag{7}$$

where $A$ is the ground truth of segmentation, while $B$ is the result by brain-map. $A \cap B$ represents the number of pixels where $A$ and $B$ overlap, and $A + B$ refers to the total number of pixels in $A$ and $B$.

Here, with referring to BrainsMapi (*Ni et al., 2020*) methods, we compared the registration/segmentation results by four registration tools at both coarse region level and fine nuclei level. As we assessed the accuracy of these methods at brain-region level, 10 brain regions, Outline, CB, CP, HB (hindbrain), HIP, HY (hypothalamus), Isocortex, MB (midbrain), OLF, and TH (thalamus), were first selected from the entire brain for comparison. Then we further picked out 50 planes in each selected brain region (totally 500 planes for 10 regions) and manually segmented them to generate the reference results (ground truth). For nuclei-level comparison, we selected nine small sub-regions, ACA, ENT, MV, PAG, RT, SSp, SUB, VISp, and VMH as targets, and performed similar operation on them with selecting five representative coronal sections for each region. To allow the manual segmentation as objective as possible, two skillful persons independently repeated the abovementioned process for five times, and a STAPLE algorithm (*Warfield et al., 2004*) was used to fuse the 10 manual segmentation results to obtain the final averaged output as the ground-truth segmentation, for each region.

## Deep neural network segmentation

### Generation of ground-truth training data

We chose 18 primary regions (levels 4 and 5): Isocortex, HPF, OLF, CTXsp, STR, PAL, CB, DORpm, DORsm, HY, MBsen, MBmot, Mbsta, P-sen, P-mot, P-sat, MY-sen, and MY-mot, in whole brain for the DNN training and performance validation. The ground-truth annotation masks for these regions were readily obtained from our bi-channel registration procedure of BIRDS. For high-generosity DNN segmentation of whole brain and incomplete brain, we specifically prepared two groups of data training and validation as following: (1) Nine whole mouse brains containing 5000 annotated sagittal slices ($660 \times 400$ for each slice) were first used as the training dataset. Then, the sagittal sections of whole brains were cropped, to generate different types of incomplete brains, as shown in *Figure 5b*. The training dataset were thus comprised both complete and incomplete sagittal sections (5000 and 4500 slices, respectively). The DNN trained by such datasets was able to infer segmentations for given complete or incomplete sagittal planes. *Figure 5—figure supplement 3* compared the DNN segmentation results of both whole brain (a) and incomplete brains (b, c, d) with the corresponding ground truths. (2) To demonstrate the performance of the DNN in segmenting sub-regions at finer scale, we chose the ground-truth images from coronal sections of specific hippocampus region (1100 slices from eight mouse brains, $570 \times 400$ for each slice) for DNN training. The corresponding ground-truth masks for the four major sub-regions of hippocampus, CA1, CA2, CA3, and the DG, were then generated by registration. We validated the DNN' performance on segmenting these small sub-regions through comparison with the ground-truths at four different coronal planes.

During network training, we defined the segmentation classes with the same number of regions, which is 19 in *Figure 5*, *Figure 5—figure supplement 3* for large incomplete (background as one class) and 5 in *Figure 5—figure supplement 2* (background as one class). Finally, each region was assigned a channel for generating the segmented output. The details of training and test datasets are shown in *Supplementary file 2*.

### Modified deeplab V3+ network

Our DNN is based on the modification of Deeplab V3+, which contains a classical encoder–decoder structure (*Chen et al., 2020*). The main framework is based on Xception, which is optimized for depthwise separable convolutions and thereby reduces the computational complexity while

maintains high performance. The network gradually reduces the spatial dimension of the feature map at the encoder stage and allows complicated information to be easily outputted at deep level. At the final stage of the encoder structure, we introduce a dilated convolution Atrous Spatial Pyramid Pooling (ASPP), which increased the receptive field of convolution by changing the stepsize of atrous. The number of cores selected in ASPP is 6, 12, and 18. To solve the issue of aliased edges in inference results of conventional Deeplab V3+, we introduced more original image information into the decoder by serially convolving the 2× and 4× down-sampling results generated at the encoder stage and concatenating them to the decoder. The network was trained for ~30,000 iterations with a learning rate of 0.001, learning momentum of 0.9 and output stride of 8. The training was implemented using two NVIDIA GeForce GTX 1080Ti graphics cards and took approximately 40 hr.

### Neuron tracing

The neurons in whole brain data were segmented semi-automatically using the Imaris software. With registering our brain to ABA, we obtained the anatomical annotation for all the segmented areas. The Autopath Mode of the Filament module was applied to trace long-distance neurons. We first assigned one point on a long-distance neuron to initiate the tracing. Then, Imaris automatically calculated the pathway in accordance with image data, reconstructed the 3D morphology, and linked it with the previous part. This procedure would repeat several times until the whole neuron, which could also be recognized by human's eye, was segmented.

### Cell counting

The Spots module and Surface module of Imaris software were used to count retrogradely labeled striatum-projecting cells in SS and VIS areas. We first separated the brain regions into multiple channels in Surface module. Then automatic creation in Spots module was applied to count cells number for each channel. To achieve accurate counting, the essential parts were the appropriate estimate of cell bodies' diameter and filtration of the chosen cells by tuning the quality parameters. The accuracy of this automatic counting procedure was compared with manual counting, which herein severed as ground truth. After obtaining the total number of cells in each brain region, according to the sub-region ranges divided by the Surface module, Imaris could also calculate the volume of each brain region. Then, with knowing the number of cell nuclei and the volume of each segmented brain region, the density of the cell nuclei inside each brain region could be calculated.

## Computing resources

We run BIRDS on a Windows 10 workstation equipped with dual Xeon E5-2630 V3 CPUs, 1 TB RAM, and two GeForce GTX 1080 Ti graphic cards. In *Supplementary file 1*, we showed the memory and time consumptions of our BIRDS plugin for processing 180 GB and 320 GB brain datasets. The image preprocessing time at stage 1 is approximately proportional to the size of data. In contrast, since the datasets for registration are down-sampled to the same size to match the Allen CCF template, the time and memory consumption at stages 2 and 3 are nearly the same for two datasets. The time and memory consumption for generating the 3D digital framework at stage 4 is proportional to the data size. It should be noted that memory with capacity at least 1.5 times of the data size is required at this step. Therefore, when applying BIRDS to larger datasets, such as rat brain, we will recommend a powerful workstation with at least one XEON CPU, 1 TB memory, and one state-of-the-art graphic card (Geforce RTX 3090), to guarantee a smooth running of whole pipeline.

## Code availability

We have made our pipeline open access for the community. We have provided source code of the BIRDS for Windows 10 at https://github.com/bleach1by1/birds_reg (*Wang, 2021a*; copy archived at swh:1:rev:22cf3d792c3887708a65ddae43d6dde7ed8b7836). FIJI plugin and ancillary installation packages are provided at https://github.com/bleach1by1/BIRDS_plugin (*Wang, 2021b*; copy archived at swh:1:rev:41e90d4518321d6ca8e806ccadb2809bfa6bd475). BIRDS contains five core modules, image preprocessing, bi-channel registration, manual correction, link with Imaris software, and deep-learning segmentation, all of which can be executed on a GUI. We also provided a step-by-step tutorial and test data to facilitate the program implementation by other researchers.

## Acknowledgements

We thank Haohong Li, Luoying Zhang, Man Jiang, Bo Xiong for discussions and comments on the work and Hao Zhang for the help on the code implementation. This work was supported by the National Key R and D program of China (2017YFA0700501 PF), the National Natural Science Foundation of China (21874052 for PF, 31871089 for YH), the Innovation Fund of WNLO (PF) and the Junior Thousand Talents Program of China (PF and YH), the FRFCU (HUST:2172019kfyXKJC077 YH).

## Additional information

### Funding

| Funder | Grant reference number | Author |
|---|---|---|
| National Natural Science Foundation of China | 21874052 | Peng Fei |
| National Natural Science Foundation of China | 31871089 | Yunyun Han |
| Innovation Fund of WNLO | | Peng Fei |
| Junior Thousand Talents Program of China | | Yunyun Han Peng Fei |
| The FRFCU | HUST:2172019kfyXKJC077 | Yunyun Han |
| National Key R&D program of China | 2017YFA0700501 | Peng Fei |
| 973 Program | 2015CB755603 | Shaoqun Zeng Yongsheng Zhang |
| Director Fund of WNLO | | Yongsheng Zhang Shaoqun Zeng |

The funders had no role in study design, data collection and interpretation, or the decision to submit the work for publication.

### Author contributions

Xuechun Wang, Data curation, Software, Validation, Visualization, Methodology, Writing - original draft; Weilin Zeng, Methodology, Software, Validation; Xiaodan Yang, Resources, Visualization; Yongsheng Zhang, Methodology, Software; Chunyu Fang, Resources; Shaoqun Zeng, Methodology; Yunyun Han, Resources, Visualization, Writing - review and editing; Peng Fei, Software, Funding acquisition, Visualization, Writing - review and editing

### Author ORCIDs

Shaoqun Zeng https://orcid.org/0000-0002-1802-337X
Peng Fei https://orcid.org/0000-0003-3764-817X

### Decision letter and Author response

Decision letter https://doi.org/10.7554/eLife.63455.sa1
Author response https://doi.org/10.7554/eLife.63455.sa2

## Additional files

### Supplementary files

• Supplementary file 1. Data size, memory cost and time consumption at different BIRDS stages for processing 180 GB STPT and 320 GB LSFM datasets.

• Supplementary file 2. Details of training dataset and test dataset for coarse and fine DNN segmentations.

• Supplementary file 3. The median values corresponding to *Figure 2d–f*.

• Transparent reporting form

## Data availability

The Allen CCF is open access and available with related tools at https://atlas.brain-map.org/. The datasets (Brain1~5) have been deposited in Dryad at https://datadryad.org/stash/share/4fes-XcJif0L2DnSj7YmjREe37yPm1bEnUiK49ELtALg and https://datadryad.org/stash/share/PWwOzH-mOqVBa_CBplDW133X5AEGwFsuoZZ4BNW_nAsQ. The code and plugin can be found at the following link: https://github.com/bleach1by1/BIRDS_plugin (copy archived at https://archive.softwareheritage.org/swh:1:rev:41e90d4518321d6ca8e806ccadb2809bfa6bd475/), https://github.com/bleach1by1/birds_reg (copy archived at https://archive.softwareheritage.org/swh:1:rev:22cf3d792c3887708a65ddae43d6dde7ed8b7836/), https://github.com/bleach1by1/birds_dl.git (copy archived at https://archive.softwareheritage.org/swh:1:rev:92d3a68c7805cbef58c834e39-c807e8cbaa902e6/), https://github.com/bleach1by1/BIRDS_demo (copy archived at https://archive.softwareheritage.org/swh:1:rev:61ad20ab070b7af9881d69df643fa4b878993f90/). All data generated or analysed during this study are included in the manuscript. Source data files have been provided for Figures 1, 2, 3, 4, 5 and Figure 2—figure supplements 3,4; Figure 5—figure supplements 2,3.

The following previously published datasets were used:

| Author(s) | Year | Dataset title | Dataset URL | Database and Identifier |
|---|---|---|---|---|
| Wang X | 2021 | brain3_4_5 | https://doi.org/10.5061/dryad.qnk98sffp | Dryad Digital Repository, 10.5061/dryad.qnk98sffp |
| Wang X | 2018 | Brain1_2 Dataset | https://doi.org/10.5061/dryad.37pvmcvj9 | Dryad Digital Repository, 10.5061/dryad.37pvmcvj9 |

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
