## [Decision Letter]

**Acceptance summary:**

A major advance in analyses of brain and neural circuit morphology is the capability to register labeled neuronal structures to a reference digital 3D brain structural atlas such as the Allen Institute for Brain Sciences mouse brain Common Coordinate Framework. In this technical advance, the authors have developed open source, automated methods for 3D morphological mapping in the mouse brain employing dual-channel microscopic neuroimage data and deep neural network learning. The authors demonstrate improved accuracy of image registration to the Allen Institute atlas and morphological reconstruction of mouse brain image sets, suggesting that the authors' approach is an important addition to the tool box for 3D brain morphological reconstruction and representation.

**Decision letter after peer review:**

Thank you for submitting your article "Bi-channel Image Registration and Deep-learning Segmentation (BIRDS) for efficient, versatile 3D mapping of mouse brain" for consideration by *eLife*. Your article has been reviewed by two peer reviewers, and the evaluation has been overseen by a Reviewing Editor and Laura Colgin as the Senior Editor. The following individuals involved in review of your submission have agreed to reveal their identity: Charles R Gerfen (Reviewer #1); Jahandar Jahanipour (Reviewer #2).

The reviewers have discussed the reviews with one another and the Reviewing Editor has drafted this decision to help you prepare a revised submission.

Summary:

Wang et al. present a technique for registering and analysis of whole mouse brain image sets to the Allen CCF. A major advance in the analysis of neuroanatomical circuits is the ability to register labeled neuronal structures in the whole mouse brain to a common mouse brain atlas. Numerous studies have used various techniques for such registration with varying success and accuracy. The authors' BIRDS process for registering whole mouse brain image sets to the Allen CCF uses a number of innovative approaches that provide some advances in the accuracy of registration, at least for the examples described. There are a number of major revisions listed below that need to be addressed.

Essential revisions:

1) Despite the steps being described in detail, parts of the image processing and registration process are difficult to understand, which requires careful revision of the text to clarify the steps in the procedure depicted in Figure 1:

– The down-sampling is straightforward. However, the "dynamic down-sampling ratios applied to the.… " is not at all clear. What exactly is being done and what is the purpose? By "dynamic down-sampling ratio" is it meant that instead of the ratio of the original (1:20 downsampling ratio to convert the original to the dimensions of the Allen CCF ) that the ratio might be different at different rostro-caudal levels of the brain? And is this to account for differences in how the brains are processed. For instance, experimental brains might have different dimensions in the x and y planes than the Allen CCF or might be deformed in different ways. And that this process makes those corrections at the appropriate rostro-caudal levels.

– The next step(s) are very difficult to follow: the "feature based iterative registration of the Allen reference image with pre-processed experimental images". This step involves the application of: 1) phase congruency algorithm to extract high contrast information from both experimental and template brain (CCF?) images and 2) enhanced axial mutual information extracted by grayscale reversal processing. And then somehow a "bi-channel" registration was performed. What is unclear is what channels are used for the "bi-channel" registration. Is a separate channel formed by the combination of the processes of 1) and 2)? Is the original image also combined with these channels? Are these processes applied to the Allen 2-photon images? And then exactly what is registered and how is that done?

– It might be better if the process steps were described separately and succinctly from comments about how this is better than other methods. The description in the Materials and methods section is somewhat better, but still confusing. The figure legend for Figure 1—figure supplement 4 comes close to being understood.

– The "bi-channel" process is the most confusing, and since it is part of the acronym should be better explained. In general, it seems straightforward, using not only the original image data but also that data that has been processed to enhance edges and other features. What is confusing is whether you are also using image data from channels that have labeled neurons and axons – which is stored in other channels?

2) Authors propose an algorithm for registration of high-resolution large-scale microscopic images of whole mouse brain datasets. This type of computational processing requires adequate memory and processing power which might not be available in a standard laboratory environment. Authors should elaborate on the computational techniques to address the large dataset handling of the proposed algorithm in regards of memory and the speed. Furthermore, is the proposed algorithm expandable to larger datasets such as super-resolution images or high-resolution whole rat brain datasets specifically in the context of memory and speed?

3) Authors discuss downsampling the dataset before registration. How does this downsampling affect the results specifically the smoothness of the borders between the regions?

4) The tracing of each fluorescently labelled neuronal cell is not clear! The method for tracing is neither explained nor referenced in the Results section.

5) Is the Deep Neural Network mainly to be used for incomplete brain datasets? Would be nice if it helped with providing better registration for registration of local structure differences, such as different layers in cortical structures or areas with multiple nuclei in addition to when there is damage to the tissue. Also, the 2-photon image data used for registration does not always provide sufficient morphologic features to identify certain brain structures, such as certain subnuclei in the brainstem and thalamus and to some extent different cortical layers. Basing the training data set on the structures detected using bi-channel procedures on 2-photon data may not best identify certain brain structures.

6) The DNN structure needs further clarification in terms of input and output. The input size is not specified; is it 2D section or 3D? How many output classes are defined? Does each region get a specific node in the output? It is not clear how the network output is selected. In "Generation of ground-truth training data" section it is mentioned that 18 primary regions were chosen for training set. In inference, the four major sub-regions of hippocampus, CA1, CA2, CA3, and the DG are not in the original 18 regions defined in the training set. The structure of the training set both in the training and test mode should be clearly explained.

7) It is recommended to report the median values of the comparison of different methods in a separate table rather than the legend of the figure for easier readability.

8) Variables in Equations 1 and 2 are not fully defined.

9) Equation 3 only explains the relationship of the final cost function to the Individual cost functions. The individual cost functions are not defined.

10) The "Marching cubes" algorithm introduced in "Generation of 3D digital map for whole brain" section is not defined. Is it a proposed algorithm by authors or an algorithm used from somewhere else? If latter, the authors should cite the paper.

---

## [Author Response]

Essential revisions:1) Despite the steps being described in detail, parts of the image processing and registration process are difficult to understand, which requires careful revision of the text to clarify the steps in the procedure depicted in Figure 1:– The down-sampling is straightforward. However, the "dynamic down-sampling ratios applied to the.… " is not at all clear. What exactly is being done and what is the purpose? By "dynamic down-sampling ratio" is it meant that instead of the ratio of the original (1:20 downsampling ratio to convert the original to the dimensions of the Allen CCF ) that the ratio might be different at different rostro-caudal levels of the brain? And is this to account for differences in how the brains are processed. For instance, experimental brains might have different dimensions in the x and y planes than the Allen CCF or might be deformed in different ways. And that this process makes those corrections at the appropriate rostro-caudal levels.

We apologize for the ambiguity brought to the reviewer. In addition to the individual difference, the sample preparation step often causes additional nonuniform deformation of samples. It means that the scaling factors of different parts of the brain are not the same, as compared to Allen CCF template. Such nonuniform deformation is found to be especially obvious along the AP axis and makes the following precise registration a lot more difficult. So at image preprocessing step, we divided the whole brain into a few z sub-stacks (6 in our demonstration) according to several selected landmark planes (7 in our demonstration) and applied appropriate re-sampling ratios, which are different, to them to finely re-adjust the depth of the sub-regions. Through compressing or stretching the depth of the sub-stacks by dynamic re-sampling ratio calculated by corresponding the positions of the landmark planes in Allen CCF3 template and sample data (Materials and methods section), we rectify the nonuniform deformation to restore a uniform rostro-caudal level of brain. This step was verified to be beneficial to the following image registration. Figure 1—figure supplement 2 of the revised manuscript shows the improvement of registration accuracy by using our dynamic downsampling preprocessing. We further illustrated this dynamic down-sampling operation in the revised Figure 1 and improved the descriptions in Results and Figure 1—figure supplement 1.

– The next step(s) are very difficult to follow: the "feature based iterative registration of the Allen reference image with pre-processed experimental images". This step involves the application of: 1) phase congruency algorithm to extract high contrast information from both experimental and template brain (CCF?) images and 2) enhanced axial mutual information extracted by grayscale reversal processing. And then somehow a "bi-channel" registration was performed. What is unclear is what channels are used for the "bi-channel" registration. Is a separate channel formed by the combination of the processes of 1) and 2)? Is the original image also combined with these channels? Are these processes applied to the Allen 2-photon images? And then exactly what is registered and how is that done?– It might be better if the process steps were described separately and succinctly from comments about how this is better than other methods. The description in the Materials and methods section is somewhat better, but still confusing. The figure legend for Figure 1—figure supplement 4 comes close to being understood.– The "bi-channel" process is the most confusing, and since it is part of the acronym should be better explained. In general, it seems straightforward, using not only the original image data but also that data that has been processed to enhance edges and other features. What is confusing is whether you are also using image data from channels that have labeled neurons and axons – which is stored in other channels?

Bi-channel is defined to be distinguished with conventional registration that merely uses the brain background images for registration. The 1^st^ primary channel consisted of brain background images of both sample and Allen CCF brains is always included in the registration. The 2^nd^ assistant channel consisted of the generated texture and geometry maps of both brains participates the registration procedure with appropriate weighting being applied. Therefore, both primary and assistant channels contain the graphic information from the background images of both sample and Allen CCF brains. Meanwhile, no neurons/axons labelled images are required at registration step. In the revised manuscript, we have further clarified the definitions and implementations of the bi-channel registration in Results and Materials and methods sections. Also, the registration procedure has been better illustrated in the revised Figure 1—figure supplement 4.

2) Authors propose an algorithm for registration of high-resolution large-scale microscopic images of whole mouse brain datasets. This type of computational processing requires adequate memory and processing power which might not be available in a standard laboratory environment. Authors should elaborate on the computational techniques to address the large dataset handling of the proposed algorithm in regards of memory and the speed. Furthermore, is the proposed algorithm expandable to larger datasets such as super-resolution images or high-resolution whole rat brain datasets specifically in the context of memory and speed?

We appreciate the reviewer’s professional advices, which are indeed helpful to the improvement of our manuscript. We have discussed the computational cost and the possible expansion to larger dataset in the Materials and methods section. A new Supplementary file 1 is also provided to summarize the data size, memory cost and time consumption at different BIRDS stages for processing 180 GB STPT and 320 GB LSFM datasets.

3) Authors discuss downsampling the dataset before registration. How does this downsampling affect the results specifically the smoothness of the borders between the regions?

In consideration of the big saving of computational costs (speed and memory), down-sampling the raw image dataset before registration is a standard operation, which shows minor effect on the smoothness of the borders between regions. This step has been widely used in many established brain registration works such as: ClearMap (Renier et al., 2016), aMAP (Niedworok et al., 2016) and MIRACL (Goubran et al., 2019) with downsampling the raw image data to 25-μm, 12.5-μm and 25-μm voxel size, respectively. In our method, we chose downsampling to 20-μm voxel size, which was within the normal range.

4) The tracing of each fluorescently labelled neuronal cell is not clear! The method for tracing is neither explained nor referenced in the Results section.

The tracing of fluorescently labelled neurons shown in Figure 3 was performed using the filament module of Imaris, which was based on the automatic neuron segmentation aided by human inspection / correction. We have added the descriptions on neuron tracing and counting in the Results and Materials and methods sections.

5) Is the Deep Neural Network mainly to be used for incomplete brain datasets? Would be nice if it helped with providing better registration for registration of local structure differences, such as different layers in cortical structures or areas with multiple nuclei in addition to when there is damage to the tissue. Also, the 2-photon image data used for registration does not always provide sufficient morphologic features to identify certain brain structures, such as certain subnuclei in the brainstem and thalamus and to some extent different cortical layers. Basing the training data set on the structures detected using bi-channel procedures on 2-photon data may not best identify certain brain structures.

We appreciate reviewer’s professional question. In our BIRDS pipeline, since the registration step has handled the whole brain datasets, DNN is designed to be used for incomplete brain datasets, which are otherwise difficult for registration. As long as finer labelling data being provided, the DNN itself certainly has the ability to segment finer brain structures, like CA1, CA2, CA3 and DG structures shown in Figure 5—figure supplement 4. Since only 2-photon and light-sheet datasets are available to us, it is a pity that we are currently unable to train the network with data containing sufficient morphologic features at certain brain structures, such as certain subnuclei in the brainstem and thalamus. However, it should be noted that BIRDS is a fully open-source approach, thus higher-quality data containing specific regions labelled from various laboratories could be processed using this approach. We also look forward to seeing increasingly stronger segmentation / capabilities shown by BIRDS in future.

6) The DNN structure needs further clarification in terms of input and output. The input size is not specified; is it 2D section or 3D? How many output classes are defined? Does each region get a specific node in the output? It is not clear how the network output is selected. In "Generation of ground-truth training data" section it is mentioned that 18 primary regions were chosen for training set. In inference, the four major sub-regions of hippocampus, CA1, CA2, CA3, and the DG are not in the original 18 regions defined in the training set. The structure of the training set both in the training and test mode should be clearly explained.

We apologize for the confusion to reviewer. The inputs for DNN are 2D image slices. We trained the network twice on dataset of 18 primary brain regions and dataset of 4 small sub-regions in hippocampus separately, to demonstrate network’s inference capability on both coarse segmentation of large incomplete brain (Figure 5) and fine segmentation of certain region of interest (Figure 5—figure supplement 4). During network training, we defined the segmentation classes the same with the number of regions, which is 19 in Figure 5 (background as one class) and 5 in Figure 5—figure supplement 4 (background as one class). Finally, each region was assigned a channel rather than a node for generating the segmented output. In the revised manuscript, we have included a Supplementary file 2 to provide the detailed information of training and test datasets. More DNN information can be found in the revised Materials and methods section and the source code we have provided.

7) It is recommended to report the median values of the comparison of different methods in a separate table rather than the legend of the figure for easier readability.

In response to reviewer’s suggestion, we have added a separate Supplementary file 3 to report the median values.

8) Variables in Equations 1 and 2 are not fully defined.

We have fully defined the variables in Equations 1 and 2 (Materials and methods section).

9) Equation 3 only explains the relationship of the final cost function to the Individual cost functions. The individual cost functions are not defined.

We have defined the individual cost functions as new Equation 4 in the Materials and methods section.

10) The "Marching cubes" algorithm introduced in "Generation of 3D digital map for whole brain" section is not defined. Is it a proposed algorithm by authors or an algorithm used from somewhere else? If latter, the authors should cite the paper.

We have cited the paper that reports Marching cubes algorithm in the revised manuscript (Lorensen and Cline, 1987).